# How should trauma discussions be approached in maternity care? Perspectives from a qualitative study with women, voluntary sector representatives and healthcare providers in the UK

Joanne Cull [iD],[1] Gillian Thomson [iD],[2] Soo Downe,[2] Michelle Fine,[3] Anastasia Topalidou [iD][2]

¹School of Nursing & Midwifery, Institute of Health & Social Care, London South Bank University, London, UK
²School of Community Health and Midwifery, University of Lancashire, Preston, UK
³The City University of New York, New York, New York, USA

**Correspondence to**
Dr Joanne Cull;
joanne.cull@lsbu.ac.uk

## ABSTRACT

**Background** Many pregnant women have a history of trauma, such as abuse or violence, which can significantly impact their mental and physical health. Discussing these experiences in maternity care presents an opportunity to support women, reduce stigma and connect them with resources. However, concerns persist about stigmatisation, re-traumatisation and unwarranted safeguarding referrals. The objective of this study was to explore how trauma discussions should be approached in maternity care, drawing on the perspectives of women with lived experience, voluntary sector representatives and healthcare providers in the UK. Findings aim to inform the development of a future intervention.

**Methods** Semistructured interviews were conducted with women with trauma histories (experts by experience; n=4), representatives of voluntary sector organisations (n=7) and healthcare providers (n=12). Reflexive thematic analysis was used to analyse the data. A qualitative content analysis approach was employed, supported by a Patient and Public Involvement and Engagement group (named as the 'Research Collective' for this study) comprising experts by experience, maternity care professionals and voluntary sector practitioners. The group contributed to both study design and data analysis.

**Findings** Five descriptive categories emerged: (1) *Rationale for discussions*—whether and why trauma should be addressed; (2) *Professionals and settings*—who should lead discussions and in what environment; (3) *Timing considerations*—when discussions should occur; (4) *Communicating about trauma*—strategies to sensitively explore prior trauma; and (5) *Supporting care providers*—training and emotional support needs. Participants highlighted both the benefits of trauma discussions and the practical, emotional and systemic challenges involved.

**Conclusion** Trauma discussions in maternity care are complex but essential. Findings provide practical, UK-specific insights into timing, communication and staff support considerations, highlighting the need for culturally sensitive, co-designed approaches to facilitate safe and effective trauma-informed care.

## STRENGTHS AND LIMITATIONS OF THIS STUDY

⇒ The study provides UK-specific qualitative insights, addressing a gap in predominantly international literature on trauma-informed perinatal practice.
⇒ A diverse sample was included, bringing together healthcare professionals, voluntary sector experts and women with lived experience of trauma.
⇒ The active involvement of a Research Collective, including women with lived experience, strengthened the study design and interpretation.
⇒ The practice-based focus of the research ensured that questions, analysis and interpretations were grounded in the realities of maternity care delivery.
⇒ The small number of women with lived experience (n=4) and the underrepresentation of some groups (eg, women with limited English proficiency and those from Asian backgrounds) may limit the breadth and transferability of findings.

## INTRODUCTION

Many pregnant women have histories of violence, trauma and abuse. For example, in the UK, a quarter of women have suffered physical, sexual or emotional abuse or witnessed domestic abuse before the age of 16, while 59% of young women in Australia report at least one adverse childhood experience.[1 2] These experiences can have profound impacts on mental and physical health, as well as health-seeking behaviours, with long-lasting consequences.[3–5] Women who have suffered trauma face heightened risk of relapses of existing mental health conditions and the onset of new disorders during the perinatal period.[6]

Discussing trauma in maternity care can help ensure effective perinatal care, identify support needs and, where appropriate,

provide information or referral to services such as mental health or substance use support.[7] Nonetheless, concerns persist regarding the potential for re-traumatisation, safeguarding referrals that women perceive as unwarranted and stigmatisation of women with traumatic histories.[8–10] Such fears often arise where disclosures of past abuse are interpreted as safeguarding risks even in the absence of current concerns, reflecting wider societal misconceptions that survivors may be unsafe parents.[11]

The objective of this study was to explore how trauma discussions should be approached in maternity care, drawing on the perspectives of women with lived experience, voluntary sector representatives and healthcare providers in the UK. To inform the development of a future intervention, we conducted qualitative interviews with these groups.

In this context, we define 'routine' as raising the issue of trauma with all women accessing maternity services, rather than selectively for those suspected of having experienced trauma.

## Patient and public involvement

The study employed critical participatory action research methodology and was supported by a Patient and Public Involvement and Engagement group (named as the 'Research Collective' for this study) comprising experts by experience, maternity care professionals and voluntary sector practitioners. The Research Collective met six times: once to review the initial doctoral application and five times during the doctoral period. The first five workshops were held online via Zoom and lasted between 1.5 hours and 3 hours; the final workshop was held in person in London and included a shared lunch.

Their involvement in framing the study, developing interview questions, selecting participants and analysing data was crucial in challenging preconceived ideas. Regular team discussions and feedback from the Research Collective upheld trustworthiness, and anonymous feedback after workshops aimed to ensure all Collective members felt heard and able to contribute.

## Reflexive note

At the outset of the study, we explored our pre-existing beliefs on routine trauma discussions and their potential impact on the study. JC and SD, midwives and maternity care researchers, shared concerns about potential harm caused by poorly conducted trauma conversations. GT, a perinatal mental health researcher, advocated for trauma-informed conversations. AT, a maternal and neonatal care researcher, stressed the importance of supportive care models. MF, an expert in critical psychology and participatory research, saw trauma as a complex source of knowledge and creativity.

As noted above, reflexivity was aided through the input of the Research Collective. Further, findings were shared at national and international conferences which gave opportunities for peer reflection. JC maintained reflexivity through journaling, supervision and counselling, further enhancing analytical rigour.

## Study design

In accordance with a critical participatory action research approach, the Research Collective played a crucial role in shaping the study methods and interview questions. Recognising the sensitivity of the topic and the importance of ensuring participant comfort and confidentiality, one-to-one interviews were chosen for data collection.

## Recruitment

Following discussions with the Research Collective, purposive sampling was undertaken to ensure representation from various maternity care professions, experience levels and demographics, as well as diverse trauma types among voluntary sector practitioners and experts by experience.

Maternity care professionals and experts from the voluntary sector (EVs) were recruited through the clinical networks of the research team and by approaching professionals and experts known to be working in this area. The recruitment of experts by experience was facilitated through collaborating with voluntary sector organisations dedicated to supporting women following trauma. These organisations distributed a recruitment flyer, inviting interested women to contact the research team directly. We aimed to respect participants' autonomy by not requiring disclosure of personal experiences to establish eligibility. Eligibility criteria included being over 18 years of age and having accessed UK maternity services at any previous time.

## Ethics

All participants received a £10 shopping voucher as a small token of appreciation, reflecting available study funds and the aim to recognise their contribution.

Recruiting experts by experience through voluntary sector organisations ensured that women did not feel coerced into participating; it also ensured that participants had access to emotional support. All recruitment discussions occurred via email, further mitigating any perceived pressure. Prior to participation, all potential participants received comprehensive study documentation, including the interview topic guide, enabling individuals to fully consider their involvement before committing.

The interview topic guide was carefully designed with the Research Collective to minimise distress among participants. Participants were not questioned about their trauma histories or personal maternity care experiences. A distress protocol was developed in case a participant should become upset during an interview. The participant information sheet stated that the interviewee could choose not to answer any question, and that they were free to stop the interview at any time and without giving a reason. Participants were also made aware that confidentiality was assured unless they disclosed that they or others were at risk of harm. The information sheet provided

details of available support services, and following the interview, participants received a debrief email reiterating those details. At the outset of the interview, participants were asked to either sign a consent form (for face-to-face interviews) or provide verbal consent (for Teams interviews).

To safeguard anonymity, each interviewee was assigned a unique code number, which was used to label all documents instead of their names. Certain demographic data were aggregated, and job titles were generalised to protect the identities of participants in unique or national roles.

### Data collection

As noted above, the Research Collective was actively engaged in shaping the interview topic guide (online supplemental material), providing invaluable insights into the content, sequence and language of the questions. The guide included questions relevant to all participants, such as *'When should maternity care providers inquire about difficult experiences?'*. Additionally, questions tailored specifically to experts from the voluntary sector (EVs) and healthcare professionals (HPs) were included, such as *'How can adequate time be allocated for these discussions?'*. A selection of trauma screening tools, identified while carrying out a systematic review which preceded this study,[11] were also used as prompts.

Two interviews were undertaken face to face, and the remainder (n=20, one of which had two participants) by Microsoft Teams. The interviews took an average of 60–90 min. A professional transcription service, with appropriate confidentiality agreements, was used to ensure accuracy. We did not set a fixed sample size in advance. Recruitment continued until sufficient depth and diversity of perspectives had been achieved. Recruitment ceased after 23 participants, as the data were rich and no new categories were emerging.

### Data analysis

We conducted a qualitative content analysis,[12] guided by our aim of developing a practical guide to support trauma discussions in maternity care. A primarily deductive approach was taken, shaped by the interview topic guide, which was informed by the literature, clinical expertise and participatory input from the Research Collective. Transcripts were read repeatedly, and meaning units relating to experiences of trauma discussions were identified, condensed and coded. Codes were then grouped into categories summarising practical aspects of trauma conversations, such as timing, setting, style of questioning, responses and provider support. The analysis was primarily manifest, attending to what participants explicitly said, but latent interpretation was also applied in later stages to highlight the underlying conditions required for safe and acceptable trauma discussions. The full research team reviewed the categories to enhance trustworthiness and mitigate personal biases. Findings were further shared and refined through two workshops with the Research Collective.

**Table 1** Participant demographic data

| | Experts by experience (n=4) | Maternity care providers and experts from the voluntary sector (n=19) |
|---|---|---|
| Sex | | |
| Female | 4 | 17 |
| Male | 0 | 2 |
| Self-described ethnic category | | |
| White, white British or white other | 3 | 15 |
| Black African or African black British | 1 | 4 |
| Age, years | | |
| 18–30 | 1 | 0 |
| 31–45 | 3 | 9 |
| 46–60 | 0 | 6 |
| Over 60 | 0 | 4 |
| Region | | |
| England | 4 | 15 |
| Wales | 0 | 2 |
| Scotland | 0 | 2 |

## RESULTS
### Participants

Of the 23 participants who took part in this study, 12 were maternity care providers, 7 were voluntary sector practitioners and 4 were experts by experience. A summary of participants' demographic data can be found in table 1, and pseudonymised job titles for maternity care professional participants in table 2. Practitioners from across a range of relevant professions were represented, including

**Table 2** Job titles—maternity care professional participants (n=12)

| | Job title |
|---|---|
| 1. | Clinical matron/specialist midwife |
| 2. | Specialist midwife for perinatal mental health |
| 3. | Consultant Liaison Psychiatrist |
| 4. | Health Visitor |
| 5. | Clinical Lead for Perinatal Mental Health |
| 6. | Clinical Lead for Perinatal Mental Health in a prison |
| 7. | Team Manager, Children's Social Care |
| 8. | Professional Midwifery Advocate |
| 9. | General Practitioner (retired) |
| 10. | Psychosexual therapist |
| 11. | Midwife—community and hospital |
| 12. | Consultant Obstetrician and Gynaecologist |

midwifery, obstetrics, perinatal mental health, health visiting, psychosexual therapy, psychiatry, children's social care and general practice. The voluntary sector practitioners specialised in supporting women after domestic abuse, birth trauma, removal of children from care, seeking asylum, sexual violence and female genital mutilation.

The categorisation of participants in the study proved more nuanced and overlapping than initially anticipated. Many maternity care professionals and voluntary sector practitioners disclosed personal or family experiences of trauma. One voluntary sector practitioner was also a qualified midwife with extensive experience supporting women in the criminal justice system. All experts by experience were actively involved in supporting women through local Maternity and Neonatal Voices Partnerships, perinatal mental health charities or expert by experience roles within mental health services. This overlap complicated classification and highlights the need for sensitivity and trauma-informed research approaches, regardless of participant groups.

## Overview of findings

Participants offered a range of insightful perspectives about trauma discussions which we organised into five descriptive categories. Due to the small number of women with lived experience of trauma (WLE; n=4), findings are presented as collated reflections across participant groups—HPs, EVs and WLEs—while highlighting divergences where they emerged. Many participants classified as HPs or EVs also disclosed personal trauma histories, further blurring participant categories. Where possible, we indicate when the perspectives of WLE differed from those of professionals.

The categories are: (1) *Rationale for discussions*, exploring whether care providers should raise the issue of previous trauma with women; (2) *Professionals and settings*, considering which professionals should carry out trauma discussions and the optimum environment; (3) *Timing considerations*, examining when trauma discussions should occur; (4) *Communicating about trauma*, describing interviewee perspectives on the style and content of trauma discussions; and (5) *Supporting care providers*, addressing the training and emotional well-being needs of professionals conducting trauma discussions. Participants are coded as HP1, HP2, etc; WLE1, WLE2, etc; and EV1, EV2, etc.

## Rationale for discussions

Participants highlighted the profound impact of sensitively addressing trauma and providing postdisclosure support in the perinatal period, with one interviewee describing it as *'an amazing opportunity and time to do this critical work'* (HP5). We heard from participants that these discussions offered an opportunity for women to reclaim agency over their prior experiences, rather than erasing traumatic memories.

They highlighted the interconnectedness of mental health and trauma experiences, suggesting that discussions surrounding these topics should be integrated. Some participants suggested a devastating link between trauma and suicide, arguing that without trauma discussions, care providers miss the opportunity to support women who may be extremely distressed. One expert by experience candidly expressed the potentially transformative impact of trauma discussions and support, commenting:

*I think it [talking about trauma and providing support] can make the difference between, it sounds dramatic but life and death. Literally. After my first birth I thought about ending my own life and this time I obviously don't anticipate that happening* (WLE1).

Care providers were seen as having an important role to play in educating women about the effects of trauma and preparing them for the possibility that the perinatal period might be challenging. Simply having the opportunity to talk was viewed as beneficial to women: *'they feel lighter, they feel like they share their burden, they feel like they can get better'* (EV4), and had the potential to greatly improve children's lives, *'interrupting that intergenerational transmission of trauma'* (HP5). Participants suggested that care providers should support both parents, with some proposing that partners should also be asked about trauma and mental health. Interviewees also underscored the potential economic benefits of implementing properly funded trauma discussions. A psychiatrist participant argued that early trauma discussions are a good investment in time to pick up problems early and ensure women are given the support they need, pointing out *'that is better for her, but it is also actually a more efficient way to run the service'* (HP3).

However, interviewees cautioned that trauma discussions will only be of value to women if they are used to improve care, rather than merely *'just asked, recorded, then ignored...'* (EV7). Broader support services, particularly mental health services, were viewed by some as inadequate, inconsistent or not trauma-informed. One HP participant cautioned that insensitive trauma conversations could be distressing and cause women to confront past experiences in an unanticipated and harmful way: *'that may not have been a big deal to them then all of a sudden oh my god that was abuse'* (HP1). Concerns were also raised across participant groups regarding the risk of distress caused by insensitive conversations and overzealous safeguarding responses:

*A good outcome is that that woman has a more positive experience of being pregnant and giving birth and the early time with her child, than she would have done without us asking. If the reality is that only 1 of 10 women who we ask has that outcome and 9 of them have disasters because suddenly social services are involved, and they are beholden to all sorts of systems and they are reporting to the police and they didn't really want to…. (EV1)*

## Professionals and settings

When exploring who should conduct trauma discussions, most experts by experience and voluntary sector practitioners felt that women would be most comfortable disclosing traumatic experiences to a female clinician, with some explicitly stating they would not disclose to a male: *'if it was me I would lie I wouldn't even open up to a man'* (EV4). While it was felt that any maternity care provider could potentially initiate trauma discussions, midwives and health visitors were particularly favoured due to their frequent contact with women during pregnancy and the postnatal period. Continuity of care was also perceived as important, enabling professionals to build rapport and create a psychologically safe environment for discussions:

> It is such a personal and very intimate part of your life, it is not something you are ready to share with a complete stranger who says 'oh hello I am your midwife, now tell me have you ever experienced trauma?' (EV6)

Nonetheless, some participants noted that even where continuity is not possible, care providers can use kindness, compassion and warmth to create a psychologically safe environment.

Participants across groups expressed concerns about discussing trauma in the clinical setting, due to its potential to inhibit disclosures. A perinatal mental health specialist midwife vividly described the lack of privacy in many clinical environments, saying her antenatal clinic was *'like Grand Central Station'* (HP2). Participants suggested that a more informal environment, with comfortable seating and refreshments, would be more conducive to sensitive discussions. For some women, clinical environments were reminiscent of previous negative experiences with statutory services. It was suggested that there should be support available for women who become upset and need space to collect themselves after the appointment, in terms of both a private space and a staff member.

## Timing considerations

All participant groups stressed the need for maternity care providers to initiate conversations about difficult experiences only when sufficient time is available to listen and respond effectively. One HP cautioned:

> If you have got 2 minutes left and you say to somebody 'so have you ever experienced sexual trauma?', no, just don't do it. Do it on a different occasion, think practically about it. Have you got the time to give the space? (HP10)

It was unanimously agreed that discussions about trauma should not occur in front of partners or young children, as their presence could inhibit open discussion. Participants also highlighted the importance of forewarning women about upcoming discussions about traumatic experiences to allow them to prepare and arrange for support if needed. Participants noted that for many women, multiple encounters are necessary before they feel safe enough to share their histories, with one emphasising the *'enormous amount of weighing up that will go on*

*before people trust and disclose'* (HP9). Some participants proposed that even where women choose not to disclose on that particular occasion, carrying out routine trauma discussions sensitively could facilitate trust and future disclosure: *'you have planted the seed of if I am ever ready I can. This is a safe space. This is a safe person'* (WLE2).

Participants acknowledged the challenges associated with discussing trauma during the first midwifery appointment due to time constraints and the predominantly closed-ended format of the appointment. One midwife participant expressed her unease at having to move on from an emotive disclosure to *'do you have a dentist, do you have a dog kind of thing'* (HP11). Participants from all population groups felt that an additional antenatal appointment specifically focused on emotional health and well-being, including discussions about traumatic experiences, would be helpful. Comments included, *'I just got goosebumps just thinking how good that would be. Yes'* (EV4); *'I think that's brilliant'* (WLE4), and *'I think it would be wonderful. And I think it would really do a lot to allay fears of women'* (EV6). A health advocate described it as a *'great idea'* (EV5) and added: *'even things that we don't share with our husbands will come out, our worries, our fears'*. They highlighted advantages such as alleviating the crowded schedule of the first maternity care appointment and providing a protected space for meaningful conversations. Participants favoured an unstructured, woman-led conversation format, with one remarking, *'no paperwork, you just go along and you hear and connect. That is really powerful'* (EV3).

## Communicating about trauma

Participants highlighted the complexities of discussing trauma, noting the need for sensitivity, clarity and accuracy. They noted that commonly used terms such as 'trauma', 'emotional abuse', 'sexual abuse' and 'physical abuse' may not resonate with women's own perceptions of their experiences, potentially hindering disclosure. For instance, one participant explained that women may not feel they have been abused *'but if you knew her history you would think she absolutely was'* (EV2). Some groups of women, such as those who are autistic or have learning disabilities, were felt to face additional challenges in understanding and articulating their experiences. Further, participants underscored the importance of developing materials with low literacy levels in mind. The use of explicit or formal language was perceived to inhibit conversations, potentially causing mothers to *'completely shut off'* (HP4). It was felt that closed-ended questions in general may deter women from disclosing because of a fear of social services involvement.

Participants universally felt that communication challenges are magnified for women with limited English proficiency, who may struggle to grasp complex information or nuances, leading to misunderstandings or embarrassment. Participants highlighted that simply translating questionnaires into a woman's first language does not guarantee her understanding as not all women are

literate in their native language. The stigma surrounding mental health in some cultures can further hinder open discussions; a midwife shared her experience of women reacting with discomfort to mental health enquiries: *'the woman will look at the partner or the granny as if to say, 'this is awful that you are even asking me this''* (HP11).

Many participants questioned the effectiveness of quantitative trauma and mental health screening tools, advocating for a relational approach instead. Views on specific screening tools varied. While some considered the Antenatal Psychosocial Risk Questionnaire to be clear and comprehensive, others questioned the utility of detailed questions such as 'when you were growing up, did you feel your mother was emotionally supportive of you?' in the absence of clear pathways for intervention or support. WLE described some questions as intrusive: *'more like child protection, you are looking if I am going to be a good mum'* (WLE3). The Kimberley Mum's Mood Scale was widely praised for its simplicity, sensitivity and visual approach, which participants felt fostered trust and honest responses. One interviewee remarked, *'it is simple but very, very effective'*. In contrast, the Adverse Childhood Experiences Questionnaire faced strong criticism for its explicit language and potential to re-traumatise women. A woman with lived experience of trauma expressed a visceral emotional reaction to the questionnaire:

*It reminds you that you weren't looked after. You weren't taken care of. You know that as a child, you weren't parented, you weren't loved in the way that a child should be loved. What upsets me isn't the act of the abuse, it is the fact that I wasn't looked after and I didn't have that love and care and what that means as an adult.* (WLE2)

Participants stressed the need for sensitive communication when women disclose trauma, advocating for active listening over intrusive questions: *'understand the difference between your own nosiness vs what is actually needed'* (HP10). Participants stressed the importance of providing independent access to support for women who choose not to disclose their experiences. They also highlighted the need for care providers to adopt a universal precautions approach, being sensitive to the possibility of trauma in every interaction:

*Treating everybody with respect and coming from a place of actually you could have had a really horrible story…'.* (WLE1)

### Supporting care providers

All participant groups highlighted the importance of training for all staff, including receptionists and clinical support, to recognise signs of possible trauma and communicate these observations to maternity care providers. Interviewees stressed the importance of recognising non-verbal cues indicating trauma or mental health struggles. Some WLEs expressed frustration at care providers' failure to notice distress: *'it was quite clear that I was distressed but they just didn't seem to realise'* (WLE1). Interpersonal skills,

centred on compassion and relationship-building, were deemed essential, although challenging to teach. Simulation with actors was suggested to enhance communication skills. Participants also considered that training in fundamental counselling skills could aid in supporting women upset during discussions of traumatic experiences.

Multiple healthcare providers talked of the burden of hearing trauma disclosures, with one describing it as *'heavy going'* (HP4), and proposed that these conversations may be particularly poignant for care providers who have endured similar experiences themselves. Participants suggested that awareness of the potential for hearing upsetting stories could mean care providers are reluctant to carry out discussions about traumatic experiences. Moreover, participants cautioned against the potential devastating consequences of interruptions and premature termination of conversations caused by provider discomfort:

*I spend so much of my time as a psychosexual therapist unpicking how clients have felt about being shut down by healthcare professionals, because they have been asked a question, but they haven't been heard and listened to […] It is very likely to be about time restrictions, or it has triggered something in this professional. But your client shouldn't have to carry that, your client just goes, oh I will never tell them again.* (HP10)

Participants emphasised the essential role of supportive management, both in managing caseloads to ensure that emotionally challenging work is distributed evenly among the team, and recognising and supporting staff who are suffering due to their own difficult life experiences. However, interviewees expressed concerns that staff might not feel comfortable disclosing trauma or subsequent mental health struggles to management due to fears of career repercussions. Further, managers themselves voiced concerns that staff support services could be seen as punitive rather than supportive. A therapist participant criticised the prevailing culture within the NHS that discourages vulnerability and prioritises stoicism, stating,

*Not wanting to be seen as weak or not able to cope, this ideology which is really strong in healthcare that you have just got to crack on with it, come on this is the job, pull up your pants, this is what you signed up for. It is not helpful, and it stops people from disclosing when the shit is hitting the fan for them.* (HP10)

The consensus among participants was that staff expected to engage in routine trauma discussions should receive regular reflexive supervision during working hours and from someone independent of the maternity team. Both group and individual supervision were deemed valuable and complementary. An expert by experience denounced the expectation for staff to conduct these discussions without proper supervision as *'completely unfair and inappropriate'* (WLE4). Stressing the importance of mandatory supervision, a therapist noted that without it, staff may not recognise the potential for burn-out or

seek support, stating, *'you don't know until you know how beneficial it is'* (HP10).

## DISCUSSION

A growing body of international literature has articulated the principles and clinical implications of trauma-informed perinatal care.[13–19] Our study was informed by this body of work and extends it by providing UK-specific, practice-oriented insights, drawing on the perspectives of maternity professionals, WLE and voluntary sector experts. By grounding the analysis in these diverse perspectives, this paper complements existing conceptual discussions and offers pragmatic considerations for when, how and by whom trauma discussions should be conducted in routine care.

While participants argued that it is crucial for maternity care providers to raise the issue of trauma, significant logistical challenges were highlighted. Interviewees noted that discussing trauma requires care, respect, cultural sensitivities, time and vulnerability. The study found that discussing traumatic experiences is a complex intervention that requires careful consideration of methodology, setting, timing, referral pathways, communication with women, and staff training and support.

The study highlighted limitations in existing tools for initiating conversations about previous trauma. While the Adverse Childhood Experiences Questionnaire was originally developed for research purposes, it is increasingly used in clinical settings, including maternity care.[8 20] Participants raised concerns about potential harms in this context, including distress for women and the risk of undermining trust between care providers and women. No currently available tool achieved widespread acceptance. The Kimberley Mum's Mood Scale, which incorporates a visual Likert Scale derived from the Edinburgh Postnatal Depression Scale and covers key well-being domains, including childhood experiences and mental health, emerged as the preferred tool among interview participants. Although participants felt this scale could be acceptable to women and facilitate open conversations, its original design for Aboriginal women in Western Australia means it would require careful adaptation for use in the UK. These findings highlight the potential value of a culturally sensitive, co-designed approach to support maternity care providers in conducting trauma-informed discussions rather than relying on a formal screening tool.

The study highlights a key challenge in current maternity care practices: broaching the topic of prior trauma during the initial booking appointment often proves ineffective and may distress women who are unprepared for such discussions. This finding, supported by feedback from interview participants, underscores the need for a separate antenatal appointment specifically focused on emotional health and well-being. This approach would provide multiple opportunities for women to address trauma-related concerns and mental health, facilitating more open and sensitive discussions. Continuity of carer was also highlighted as crucial for building the trust necessary for these sensitive conversations.

Implementing such approaches, however, must contend with practical constraints. Maternity services remain overstretched and understaffed, with performance-driven priorities and limited investment in preventative care.[21 22] Despite substantial investment in perinatal mental health in the UK,[22] these services remain under-resourced and sustainability is a concern, particularly in resource-limited NHS trusts. There is also a scarcity of perinatal-specific psychological therapies and limited evidence regarding the effectiveness of currently recommended interventions.[23] Alongside advocacy for expanded mental health provision, enhanced staff training, including simulation-based programmes such as the Perinatal Interprofessional Psychosocial Education programme for Maternity Clinicians,[24] may help maternity care professionals conduct sensitive psychosocial and trauma-related conversations while wider systemic improvements are developed.

The study explores the emotional challenge care providers face when hearing disclosures of traumatic experiences, especially for those who have experienced trauma themselves. Survivors of trauma may find that supporting women who have faced similar experiences can evoke distressing memories.[25] It is essential that staff involved in trauma discussions be provided with independent, professional support services. Staff may be reluctant to share personal experiences with colleagues they know or may not recognise when their stress and burnout levels are escalating. A fundamental culture shift is necessary to ensure staff access available support. This transition necessitates moving beyond merely 'offering' support to those in need, towards integrating support as a routine aspect of daily work life, actively provided to all staff during working hours. Such a cultural transformation has the potential to significantly improve the effectiveness of support initiatives.[26]

Taken together, these findings situate UK maternity practice within the broader international discourse on trauma-informed care, while highlighting context-specific, practical considerations for implementing sensitive and effective trauma discussions in routine care. Drawing on insights from the interviews and a previous systematic review and qualitative synthesis,[11] we have developed a set of guiding principles to inform the design and delivery of routine trauma discussions in maternity care. These principles emphasise staff training, emotional support, woman-centred communication and culturally inclusive practices. The development and evaluation of this approach will be reported in a subsequent publication.

### Strengths and limitations of the study

This study provides UK-specific, practice-oriented insights into trauma discussions in maternity care, complementing existing international literature on trauma-informed perinatal practice.[13–19] By including a diverse range of

healthcare professionals, experts from the voluntary sector, and women with lived experience of trauma—the study captures multiple perspectives on the practicalities, challenges and perceived benefits of trauma discussions. These insights highlight considerations such as timing, setting, communication approaches and staff support that are essential for implementing trauma-informed care in UK maternity services. While the study does not provide a prescriptive framework for implementation, it identifies key pragmatic issues that must be addressed to make trauma-informed discussions feasible and acceptable in practice.

One of the study's key strengths lies in the active engagement of the Research Collective, including women with lived experience and voluntary sector experts, throughout the research process. This collaboration ensured that findings reflect not only professional perspectives but also experiential insights, increasing their relevance for real-world practice.

Limitations include the small number of women with lived experience (n=4), meaning findings cannot be assumed to reflect the full diversity of this group's perspectives. Many participants from professional and voluntary sector backgrounds disclosed personal trauma histories, which further blurs the boundaries between participant categories. While we have highlighted divergences where they were apparent and explicitly avoided presenting professional perspectives as universally endorsed by women with lived experience, it is possible that some nuances specific to women's experiences are underrepresented. Future research should aim to include larger numbers of participants with lived experience to explore in greater depth potential points of disagreement or unique insights that may not align with professional perspectives. Purposive sampling and the exclusion of some underrepresented groups, including women with limited English proficiency and those from Asian backgrounds, may also limit generalisability.

## CONCLUSION

This study presents findings from semistructured interviews with a range of expert stakeholders. Key findings include the importance of time, adequate support for staff and effective referral mechanisms to enable maternity care providers to initiate discussions about trauma with women, and to ensure appropriate personalised follow-up in the event of disclosure. Central to these conversations are trust and relationships, necessitating careful consideration in care provision. Furthermore, the research underscores the necessity of comprehensive staff training and support.

The interview findings illuminate critical aspects of trauma-informed perinatal care, offering valuable insights into the complexities and challenges faced by both women and maternity care providers. By focusing on routine trauma discussions during the perinatal period, the research addresses a significant gap in existing literature and provides practical solutions to improve the quality of care for women who have experienced trauma.

Given the current resource constraints, a pragmatic next step could be to pilot routine trauma discussions in a well-resourced maternity service, with investment in staff training, supervision and an additional appointment dedicated to emotional health. Lessons from such a pilot could then inform whether, and how, routine trauma identification might be safely and effectively introduced more broadly.

**Acknowledgements** We thank the EMPATHY study Research Collective for their invaluable contributions to this study: Laura Abbott, Juliet Albert, Kirsty Armstrong, Jill Benjoya Miller, Ang Broadbridge, Emma Brooks, Geraldine Butcher, Jo Doherty, Amber Jackson, Isobel Martin, Elsa Montgomery, Sam Pointon, Sarah-Jayne Pomeroy, Erjola Sadria, Gill Skene, Memuna Sowe, Kim Thomas and Lucy Warwick-Guasp.

**Contributors** JC (guarantor): Conceptualisation, formal analysis, investigation, methodology, writing of the original draft, writing of the review and editing. GT: Formal analysis, methodology, supervision, writing of the review and editing. SD: Supervision, methodology, writing of the review and editing. MF: Supervision, methodology, writing of the review and editing. AT: Supervision, methodology, writing of the review and editing.

**Funding** JC was funded by a National Institute for Health Research (NIHR) Wellbeing of Women Doctoral Fellowship for this work (grant number NIHR301525). This paper presents independent research funded by the NIHR and the charity Wellbeing of Women. The views expressed are those of the authors and not necessarily those of Wellbeing of Women, the NHS, the NIHR or the Department of Health and Social Care. The funders had no role in study design, data collection and analysis, decision to publish, or preparation of the manuscript. https://www.nihr.ac.uk/, https://www.wellbeingofwomen.org.uk/.

**Competing interests** None declared.

**Patient and public involvement** Patients and/or the public were involved in the design, or conduct, or reporting, or dissemination plans of this research. Refer to the Methods section for further details.

**Patient consent for publication** Not applicable.

**Ethics approval** This study involves human participants. Ethics approval was obtained from the University of Central Lancashire Health Ethics Review Panel (reference HEALTH 0220). Participants gave informed consent to participate in the study before taking part.

**Provenance and peer review** Not commissioned; externally peer reviewed.

**Data availability statement** Data are available upon reasonable request. The qualitative data that support the findings of this study are not publicly available due to their sensitive nature and to protect participant confidentiality. De-identified data are available from the corresponding author on reasonable request.

**ORCID iDs**
Joanne Cull https://orcid.org/0000-0001-8990-154X
Gillian Thomson https://orcid.org/0000-0003-3392-8182
Anastasia Topalidou https://orcid.org/0000-0003-0280-6801

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
