## [Reviewer comments · BMJ Open]

ARTICLE DETAILS

Title (Provisional)

How should trauma discussions be approached in maternity care? Perspectives from a qualitative study with women, voluntary sector representatives, and healthcare providers in the United Kingdom

Authors

Cull, Joanne; Thomson, Gillian; Downe, Soo; Fine, Michelle; Topalidou, Anastasia

VERSION 1 - REVIEW

Reviewer	1
Name	Sved Williams, Anne
Affiliation	University of Adelaide, Department of Psychiatry
Date	21-Mar-2025
COI	None

Review "routine discussion of trauma" BMJ Open

This is very good qualitative research to address the topic that the authors have chosen to study. The steps to identify a broad group to interview, the questions asked, the themes identified and some of the conclusions are all very well and carefully outlined and discussed.

In the introduction and in the discussion there is some focus around the needs for pathways to care, clarifying that for most women, simply identifying the pathway isn't sufficient but some sort of counselling or therapy would be appropriate. However, in most countries, not only are services generally unavailable, but to the best of my knowledge there is little evidence of what therapies are likely to be beneficial even where perinatal mental health services are better staffed.

The study found that in some cases, the women found it helpful and perhaps freeing to be asked the questions about whether they had experienced trauma, whether by questionnaire or interview, even without referral on but I would assume that for most, trauma identification would feel like the first step in a long pathway that some might be prepared to undertake. Providing evidence-based therapies for perinatal women for trauma and also training and support for relevant staff is a great goal but given the frequency of trauma

experiences it seems a long way off. It did make me wonder whether first steps would be advocacy for better staffing for perinatal mental health services (very successful of course in UK), and increased training of midwives and other health staff using a package such as PIPE-MC (not my program!) as well as a further search for evidence-based therapies for trauma.

In other words, whilst the study is good, I think the discussion and conclusions do not go far enough. Routine teaching of trauma-informed care and looking for evidence-based therapies which could be implemented need to be ..reachable/at least discussed and I was thinking of a conclusion/recommendation for at least one service in the UK where routine questioning re trauma is implemented for a period of time and looking to see how it works out in a very well-resourced service (staff training and supervision, extra time for that second routine interview, development of a questionnaire – a long way to go) before routine identification would be introduced elsewhere.

Reviewer	2
Name	Isobel, S
Affiliation	The University of Sydney
Date	09-Jul-2025
COI	None

Thank you for the opportunity to review this qualitative study around routine conversations about trauma in the perinatal period. This is an important topic and strengthened by principles of participatory action research. I have some areas of concern around the combined analysis of providers and patients into a singular authoritative voice, the analysis method (and the themes) and the lack of positioning in the context of what is known about trauma informed practice.

Comments:

In the part of the statement of significance entitled 'Problem', I think 'the problem' is both what you identify about long term impacts but also about the relevance of trauma to the perinatal period itself.

At times you mention 'previous trauma' but it is my understanding that while the events were previous, the effects are often current. It makes sense to talk about 'previous' when you are referring to prevalence of exposure and also when trying to distinguish from trauma occurring during the perinatal period itself but there are other times it could be reworded. For example in introduction first para 'Women who have suffered previous trauma' could be phrased as 'Women who have experiences of trauma'

In intro second para you start with 'Discussing previous trauma within maternity care offers an opportunity to inform pregnant women about its potential impact and guide them towards relevant support such as substance abuse treatment or mental health services' - I

am a bit concerned by the primary focus being on educating and referral, these things are important but I think another key reasons for discussing it is to ensure the delivery of effective perinatal care and identify support needs during this period.

In the second para of intro where you say 'unwarranted safeguarding referrals' it would be good to explain this a little more as I am not sure what it really means. Who decides if they are warranted or not? I assume staff are following mandatory reporting requirements.

Thank you for including the reflexive note- this is an important inclusion. The first sentence feels a bit incomplete- did you commence by doing this or when and how did this occur.

Pg 8 you refer to workshops with the collective but these have not yet been mentioned- what workshops?

I am surprised that participants only received a £10 'thank you' voucher. Is there any reference or justification you can give for this very small reimbursement for people with lived experience? Perhaps even just 'in line with ethical guidance, Research Collective consultation and available funds...'

The discovery that the categories of professionals and people with LE is a very important point- thank you for addressing this. I am not sure what you mean by 'sensitivity and flexibility in research classification' though? Classification of what? I think it does highlight a need for sensitivity and trauma informed research approaches, regardless of participant groups.

In the findings, I am not convinced that your themes are themes- as they define aspects of the process I wonder if they are more summarising categories or a form of synthesis. I know this has significant methodological implications but it is the most common error in thematic analysis (please see this paper by Braun and Clarke discussing this: Braun, V., & Clarke, V. (2023). Toward good practice in thematic analysis: Avoiding common problems and being a knowing researcher. *International journal of transgender health*, 24(1), 1-6.). You may need to redefine your methods.

While I take your point that there is a blurring of participant roles, it is tricky to see the findings presented as collective experiences. What health professionals think is beneficial is known to not always align to what patients think is beneficial, especially in the field of trauma. For example, I think we need to be sure that women want to be educated, just as much as HPs see it as useful. As such, I am unsure about the decision to report findings in such a collated way. You state your findings as facts but you are combining the voices of professionals and women in a way that makes me a bit uneasy. I spent awhile thinking about this point because I don't want to be unhelpful and obviously with 4 LE participants, it is hard to separate out your findings but I do wonder about needing to either address this much more explicitly in methods and limitations, but I also wonder if there is a way to bring in a bit more equipoise about what professionals thought and how women endorsed these ideas or not by separating out ideas in the findings a bit. Currently the findings read to me as

helpful and interesting feedback from health professionals. The risk is that the voices of 4 women are used to endorse health professionals perspectives. Were there any points they didn't agree with? Did the women suggest it was hard for staff?

In discussion

Suggest the 'multi-cultural sensitivities' could just be called cultural sensitivities.

The discussion of a framework that you have developed but aren't presenting here should be removed or moved right down to an implication.

Important to acknowledge that the ACE questionnaire is intended for research not clinical purposes. It would be inappropriate to use it clinically.

I am not sure about the statement 'Consequently, there is a pressing need for a culturally sensitive, co-designed tool for use in maternity services.' As you have largely presented findings that suggest a tool may not be helpful. Perhaps you could cushion this statement a bit- there is a possible need for (rather than a pressing need) and say also what purpose the tool would serve- do you mean a screening tool or a tool to support midwives?

Your discussion does not reference any of the contemporary literature about trauma informed perinatal practice... for example (just to name a few):

- Sachdeva, J., Yang, S. N., Gopalan, P., Worley, L. L., Mittal, L., Shirvani, N., ... & Byatt, N. (2022). Trauma informed care in the obstetric setting and role of the perinatal psychiatrist: A comprehensive review of the literature. *Journal of the Academy of Consultation-Liaison Psychiatry, 63*(5), 485-496.
- Sperlich, M., Seng, J. S., Li, Y., Taylor, J., & Bradbury-Jones, C. (2017). Integrating trauma-informed care into maternity care practice: conceptual and practical issues. *Journal of midwifery & women's health, 62*(6), 661-672.
- Kuzma, E. K., Pardee, M., & Morgan, A. (2020). Implementing patient-centered trauma-informed care for the perinatal nurse. *The Journal of Perinatal & Neonatal Nursing, 34*(4), E23-E31.
- Delap, N. (2021). Trauma-informed care of perinatal women. *Complex Social Issues and the Perinatal Woman, 15*-33.
- Gerber, M. R. (2019). Trauma-informed maternity care. *Trauma-informed healthcare approaches: A guide for primary care, 145*-155.
- Owens, L., Terrell, S., Low, L. K., Loder, C., Rhizal, D., Scheiman, L., & Seng, J. (2022). Universal precautions: the case for consistently trauma-informed reproductive healthcare. *American journal of obstetrics and gynecology, 226*(5), 671-677.
- Nagle-Yang, S., Sachdeva, J., Zhao, L. X., Shenai, N., Shirvani, N., Worley, L. L., ... & Byatt, N. (2022). Trauma-informed care for obstetric and gynecologic settings. *Maternal and Child Health Journal, 26*(12), 2362-2369.

- Isobel, S. (2023). Trauma and the perinatal period: A review of the theory and practice of trauma-sensitive interactions for nurses and midwives. *Nursing Open*, 10(12), 7585-7595.

In strengths and limitations: Can you be clearer how your study addresses the need for practical implementation of trauma-informed perinatal care? I think your study informs the need for this but doesn't guide how to do it or what it means (see all the literature above which addresses this)

Limitations should talk about the small amount of folks with LE and also the collated approach to experiences .

VERSION 1 - AUTHOR RESPONSE

Reviewer comment	Response
Reviewer 1 comments: In the introduction and in the discussion there is some focus around the needs for pathways to care, clarifying that for most women, simply identifying the pathway isn't sufficient but some sort of counselling or therapy would be appropriate. However, in most countries, not only are services generally unavailable, but to the best of my knowledge there is little evidence of what therapies are likely to be beneficial even where perinatal mental health services are better staffed. The study found that in some cases, the women found it helpful and perhaps freeing to be asked the questions about whether they had experienced trauma, whether by questionnaire or interview, even without referral on but I would assume that for most, trauma identification would feel like the first step in a long pathway that some might be prepared to undertake. Providing evidence-based therapies for perinatal women for trauma and also training and support for relevant staff is a great goal but given the frequency of trauma experiences it seems a long way off. It did make me wonder whether first steps would be advocacy for better staffing for perinatal mental health services (very successful of course in UK), and increased training of midwives and other health staff using a package such as PIPE-MC (not my program!)	Thank you for your thoughtful and helpful comments. We have added the following text to the discussion section: Implementing such approaches, however, must contend with practical constraints. Maternity services remain overstretched and understaffed, with performance-driven priorities and limited investment in preventative care (Darzi, 2024; NHS, 2019). Despite substantial investment in perinatal mental health in the UK (NHS, 2019), these services remain under-resourced, and sustainability is a concern, particularly in resource-limited NHS trusts. There is also a scarcity of perinatal-specific psychological therapies and limited evidence regarding the effectiveness of currently recommended interventions (Jones et al., 2023). Alongside advocacy for expanded mental health provision, enhanced staff training, including simulation-based programmes such as the Perinatal Interprofessional Psychosocial Education program for Maternity Clinicians (PIPE-MC; Schmied et al., 2023) may help maternity care professionals conduct sensitive psychosocial and trauma-related conversations while wider systemic improvements are developed. We have added this text to the conclusion section: Given the current resource constraints, a pragmatic next step could be to pilot routine trauma discussions in a well-resourced maternity service, with investment in staff training, supervision, and an additional appointment dedicated to emotional health. Lessons from such a pilot could then inform whether, and how, routine trauma identification might be safely and effectively introduced more broadly.

as well as a further search for evidence-based therapies for trauma. In other words, whilst the study is good, I think the discussion and conclusions do not go far enough. Routine teaching of trauma-informed care and looking for evidence-based therapies which could be implemented need to be ..reachable/at least discussed and I was thinking of a conclusion/recommendation for at least one service in the UK where routine questioning re trauma is implemented for a period of time and looking to see how it works out in a very well-resourced service (staff training and supervision, extra time for that second routine interview, development of a questionnaire – a long way to go) before routine identification would be introduced elsewhere.	
Reviewer 2 comments: In the part of the statement of significance entitled ‘Problem’, I think ‘the problem’ is both what you identify about long term impacts but also about the relevance of trauma to the perinatal period itself.	Thank you for taking the time to review the manuscript so thoroughly and for your insightful comments, which have strengthened the paper. Thank you for this helpful observation. The statement of significance has been removed from the revised manuscript at the request of the editor, so this comment is no longer applicable. We have therefore not made further changes in relation to this point.
At times you mention ‘previous trauma’ but it is my understanding that while the events were previous, the effects are often current. It makes sense to talk about ‘previous’ when you are referring to prevalence of exposure and also when trying to distinguish from trauma occurring during the perinatal period itself but there are other times it could be reworded. For example in introduction first para ‘Women who have suffered previous trauma’ could be phrased as ‘Women who have experiences of trauma’.	We agree that the wording sometimes implied trauma was purely in the past, rather than ongoing in its effects. We have revised throughout (e.g. “traumatic experiences” rather than “previous trauma”) except where “previous trauma” refers specifically to prevalence or to distinguish from trauma arising in the perinatal period.
In intro second para you start with ‘Discussing previous trauma within maternity care offers an opportunity to inform pregnant women about its potential impact and guide them towards relevant support such as substance abuse treatment or mental health services’- I am a bit	We have revised this sentence to emphasise that trauma discussions are also critical to delivering safe and effective perinatal care and identifying support needs, not only for education or referral.

concerned by the primary focus being on educating and referral, these things are important but I think another key reasons for discussing it is to ensure the delivery of effective perinatal care and identify support needs during this period.	Discussing trauma in maternity care can help ensure effective perinatal care, identify support needs, and, where appropriate, provide information or referral to services such as mental health or substance use support (Flanagan et al., 2018).
In the second para of intro where you say 'unwarranted safeguarding referrals' it would be good to explain this a little more as I am not sure what it really means. Who decides if they are warranted or not? I assume staff are following mandatory reporting requirements.	We appreciate the opportunity to clarify what we meant by 'unwarranted safeguarding referrals'. We agree that mandatory reporting is required in some cases, such as female genital mutilation. However, as our earlier systematic review demonstrated (Cull et al., 2023), many women fear that disclosure of past trauma will trigger referrals they see as unnecessary, particularly where there is no immediate safeguarding risk. Women in those studies described concerns that their histories of abuse were sometimes used to assess their potential as parents rather than to identify support needs, and cited misconceptions such as 'the abused become abusers' as driving these fears. We have revised the text as follows: Nonetheless, concerns persist regarding the potential for re-traumatisation, unwarranted safeguarding referrals that women perceive as unwarranted, and stigmatisation of those with traumatic histories (Ford et al., 2019; Underwood, 2020; Racine, Killam & Madigan, 2020). Such fears often arise where disclosures of past abuse are interpreted as safeguarding risks even in the absence of current concerns, reflecting wider societal misconceptions that survivors may be unsafe parents (Cull et al., 2023).
Thank you for including the reflexive note- this is an important inclusion. The first sentence feels a bit incomplete- did you commence by doing this or when and how did this occur.	We have clarified that reflexive exploration of our pre-existing beliefs occurred at the outset of the study, as shown in the revised opening sentence of the reflexive note: At the outset of the study, we explored our pre-existing beliefs on routine trauma discussions and their potential impact on the research.
Pg 8 you refer to workshops with the collective but these have not yet been mentioned- what workshops?	We have clarified the reference to workshops as detailed below. The Research Collective met six times: once to review the initial doctoral application and five times during the doctoral period. The first five workshops were held online via Zoom

	and lasted between 1.5 and 3 hours; the final workshop was held in person in London and included a shared lunch.
I am surprised that participants only received a £10 'thank you' voucher. Is there any reference or justification you can give for this very small reimbursement for people with lived experience? Perhaps even just 'in line with ethical guidance, Research Collective consultation and available funds...'	We thank the reviewer for this comment. The £10 voucher was intended as a modest token of appreciation for participants' time and contribution, reflecting the study's available funds. We have added a statement to justify this: All participants received a £10 shopping voucher as a small token of appreciation, reflecting available study funds and the aim to recognise their contribution.
The discovery that the categories of professionals and people with LE is a very important point- thank you for addressing this. I am not sure what you mean by 'sensitivity and flexibility in research classification' though? Classification of what? I think it does highlight a need for sensitivity and trauma informed research approaches, regardless of participant groups.	We agree with your clarification and have amended the text accordingly: This overlap complicated classification and highlights the need for sensitivity and trauma-informed research approaches, regardless of participant groups.
In the findings, I am not convinced that your themes are themes- as they define aspects of the process I wonder if they are more summarising categories or a form of synthesis. I know this has significant methodological implications but it is the most common error in thematic analysis (please see this paper by Braun and Clarke discussing this: Braun, V., & Clarke, V. (2023). Toward good practice in thematic analysis: Avoiding common problems and being a knowing researcher. International journal of transgender health, 24(1), 1-6.). You may need to redefine your methods.	We are grateful for the reviewer's comment regarding our use of themes. On reflection, we agree that our findings are more appropriately described as categories that summarise key aspects of the process, rather than interpretive themes in the sense intended by reflexive thematic analysis. We have therefore redefined our approach as qualitative content analysis, primarily at the manifest level with some latent interpretation in developing the final framework. The methods and findings sections have been revised accordingly. Methods: We conducted a qualitative content analysis (Bengtsson, 2016), guided by our aim of developing a practical guide to support trauma discussions in maternity care. A primarily deductive approach was taken, shaped by the interview topic guide, which was informed by the literature, clinical expertise, and participatory input from the Research Collective. Transcripts were read repeatedly, and meaning units relating to experiences of trauma discussions were identified, condensed, and coded. Codes were then grouped into categories summarising practical aspects of trauma conversations, such as timing, setting, style of questioning, responses, and provider support. The analysis was primarily

	manifest, attending to what participants explicitly said, but latent interpretation was also applied in later stages to highlight the underlying conditions required for safe and acceptable trauma discussions. The full research team reviewed the categories to enhance trustworthiness and mitigate personal biases. Findings were further shared and refined through two workshops with the Research Collective. Findings: Participants offered a range of insightful perspectives about trauma discussions which we organised into five descriptive categories.
While I take your point that there is a blurring of participant roles, it is tricky to see the findings presented as collective experiences. What health professionals think is beneficial is known to not always align to what patients think is beneficial, especially in the field of trauma. For example, I think we need to be sure that women want to be educated, just as much as HPs see it as useful. As such, I am unsure about the decision to report findings in such a collated way. You state your findings as facts but you are combining the voices of professionals and women in a way that makes me a bit uneasy. I spent awhile thinking about this point because I don't want to be unhelpful and obviously with 4 LE participants, it is hard to separate out your findings but I do wonder about needing to either address this much more explicitly in methods and limitations, but I also wonder if there is a way to bring in a bit more equipoise about what professionals thought and how women endorsed these ideas or not by separating out ideas in the findings a bit. Currently the findings read to me as helpful and interesting feedback from health professionals. The risk is that the voices of 4 women are used to endorse health professionals perspectives. Were there any points they didn't agree with? Did the women suggest it was hard for staff?	We acknowledge this limitation. We have added to the methods and limitations section that combining data across groups risks obscuring differences in perspectives. Where possible, we have clarified whether themes were shared across groups, and we have highlighted where women's perspectives diverged or raised distinct concerns. We now note explicitly that the voices of women with lived experience are fewer in number and should be interpreted with caution.
Suggest the 'multi-cultural sensitivities' could just be called cultural sensitivities.	Changed as suggested.

The discussion of a framework that you have developed but aren't presenting here should be removed or moved right down to an implication.	We have moved mention of the framework to the bottom of the discussion section and rephrased for clarity as follows: Drawing on insights from the interviews and a previous systematic review and qualitative synthesis (Cull et al., 2023), we have developed a set of guiding principles to inform the design and delivery of routine trauma discussions in maternity care. These principles emphasise staff training, emotional support, woman-centred communication, and culturally inclusive practices. The development and evaluation of this approach will be reported in a subsequent publication.
Important to acknowledge that the ACE questionnaire is intended for research not clinical purposes. It would be inappropriate to use it clinically.	We have amended the text as follows: While the ACE questionnaire was originally developed for research purposes, it is increasingly used in clinical settings, including maternity care (Ford et al., 2019; Hardcastle & Bellis, 2019). Participants raised concerns about potential harms in this context, including distress for women and the risk of undermining trust between care providers and women.
I am not sure about the statement 'Consequently, there is a pressing need for a culturally sensitive, co-designed tool for use in maternity services.' As you have largely presented findings that suggest a tool may not be helpful. Perhaps you could cushion this statement a bit- there is a possible need for (rather than a pressing need) and say also what purpose the tool would serve- do you mean a screening tool or a tool to support midwives?	We have revised the text as follows: These findings highlight the potential value of a culturally sensitive, co-designed approach to support maternity care providers in conducting trauma-informed discussions, rather than relying on a formal screening tool
Your discussion does not reference any of the contemporary literature about trauma informed perinatal practice	Thank you for highlighting the need to engage more fully with the contemporary literature on trauma-informed perinatal care. We were aware of these contributions and have followed this literature closely, including discussions with several of the authors. We agree that it is important to situate our findings in relation to this work and have therefore revised the opening of the discussion to reference and acknowledge recent reviews and conceptual discussions (Sperlich et al., 2017; Gerber, 2019; Kuzma et al., 2020; Delap, 2021; Sachdeva et al., 2022; Owens et al., 2022; Nagle-Yang et al., 2022; Isobel, 2023).

	In line with this body of research, our study reinforces the value of trauma-informed approaches, but we position our contribution as UK-specific and practice-oriented, drawing on interviews with maternity care professionals, voluntary sector experts, and women with lived experience. This situates our paper as complementary to the broader conceptual literature, while offering pragmatic insights into how trauma discussions are currently managed in routine practice and where challenges remain.
In strengths and limitations: Can you be clearer how your study addresses the need for practical implementation of trauma-informed perinatal care? I think your study informs the need for this but doesn't guide how to do it or what it means (see all the literature above which addresses this)	Thank you for this helpful suggestion. We have revised the strengths and limitations section to clarify how the study informs practical implementation of trauma-informed perinatal care. Specifically, we now highlight that the study provides UK-specific, practice-oriented insights into the timing, setting, communication approaches, and staff support considerations that are essential for trauma-informed discussions. While we do not present a prescriptive framework in this paper, the findings identify key pragmatic issues that must be addressed to make trauma-informed care feasible and acceptable in practice.
Limitations should talk about the small amount of folks with LE and also the collated approach to experiences	Now included.